# Deep Classifiers with Label Noise Modeling and Distance Awareness

**Vincent Fortuin**[*][†]                                                                                      *vbf21@cam.ac.uk*
*Department of Computer Science, ETH Zürich*
*Department of Engineering, University of Cambridge*

**Mark Collier**[†]                                                                                      *markcollier@google.com*
*Google Research*

**Florian Wenzel, James Allingham, Jeremiah Liu, Dustin Tran, Balaji Lakshminarayanan,
Jesse Berent, Rodolphe Jenatton, Effrosyni Kokiopoulou**
*Google Research*

**Reviewed on OpenReview:** *https://openreview.net/forum?id=Id7hTt78FV*

## Abstract

Uncertainty estimation in deep learning has recently emerged as a crucial area of interest to advance reliability and robustness in safety-critical applications. While there have been many proposed methods that either focus on distance-aware model uncertainties for out-of-distribution detection or on input-dependent label uncertainties for in-distribution calibration, both of these types of uncertainty are often necessary. In this work, we propose the HetSNGP method for jointly modeling the model and data uncertainty. We show that our proposed model affords a favorable combination between these two types of uncertainty and thus outperforms the baseline methods on some challenging out-of-distribution datasets, including CIFAR-100C, ImageNet-C, and ImageNet-A. Moreover, we propose HetSNGP Ensemble, an ensembled version of our method which additionally models uncertainty over the network parameters and outperforms other ensemble baselines.

## 1 Introduction

While deep learning has led to impressive advances in predictive accuracy, models often still suffer from overconfidence and ill-calibrated uncertainties (Ovadia et al., 2019). This is particularly problematic in safety-critical applications (e.g., healthcare, autonomous driving), where uncertainty estimation is crucial to ensure reliability and robustness (Filos et al., 2019; Dusenberry et al., 2020).

Predictive uncertainties generally come in two flavors: *model uncertainty* (also known as epistemic) and *data uncertainty* (also known as aleatoric) (Murphy, 2012). Model uncertainty measures how confident the model should be based on what it knows about the world, that is, how much it can know about certain test data given the training data it has seen. Data uncertainty measures the uncertainty intrinsic in the data itself, for example due to fundamental noise in the labelling process. Good model uncertainty estimates allow for out-of-distribution (OOD) detection, that is, for recognizing data examples that are substantially different from the training data. On the other hand, good data uncertainty estimates allow for in-distribution calibration, that is, knowing which training (or testing) data examples the model should be more or less confident about. Fig. 1 demonstrates model uncertainty for a synthetic dataset, while Fig. 2 shows the effect of modeling the data uncertainty arising from noisy labels. Ideally we would like a single model which offers in-distribution uncertainty modeling while reverting to uncertain predictions when faced with OOD examples.

---

[*]The research was done during an internship at Google Research.
[†]Equal contribution.

Many recently proposed models for uncertainty estimation excel at one or the other of these uncertainty types. For instance, the spectral-normalized Gaussian process (SNGP) (Liu et al., 2020) uses a latent Gaussian process to achieve distance-aware model uncertainties and thus affords excellent OOD detection. Conversely, the heteroscedastic classification method (Collier et al., 2020; 2021) offers superb in-distribution calibration and accuracy by modeling input- and class-dependent label noise in the training data. However, there have been few attempts to combine the complementary benefits of these two types of uncertainty modeling (e.g., Depeweg et al., 2018) (see related work in Section 4).

In this work, we propose the heteroscedastic SNGP (HetSNGP), which allows for joint modeling of model and data uncertainties using a hierarchical latent variable model. We show that HetSNGP gives good in-distribution and OOD accuracy and calibration, yielding a model with uncertainties suitable for deployment in critical applications.

Our main contributions are:

- We propose a new model, the heteroscedastic spectral-normalized Gaussian process (HetSNGP), with both distance-aware model and data uncertainties.

- We describe an efficient approximate inference scheme that allows training HetSNGP with a computational budget comparable to standard neural network training.

- We introduce a new large-scale OOD benchmark based on ImageNet-21k. We hope this benchmark will prove useful for future research in the field.

- We show empirically on different benchmark datasets that HetSNGP offers a favorable combination of model and data uncertainty. It generally preserves the SNGP's OOD performance and the heteroscedastic in-distribution performance. It even outperforms these baselines on some datasets, where both OOD and heteroscedastic uncertainties are helpful.

- We propose an ensembled version of our model, the HetSNGP Ensemble, which additionally accounts for uncertainty over the model parameters and outperforms other ensemble baselines.

## 2 Background

### 2.1 Model uncertainty and spectral-normalized Gaussian process

Model uncertainty (or epistemic uncertainty) captures all the uncertainty about whether a given model is correctly specified for a certain task, given the training data. Before any training data has been observed, this uncertainty depends only on the prior knowledge about the task, which can for instance be encoded into distributions over the model parameters or into the architecture of the model (i.e., the model class) (Fortuin, 2021). After training data has been observed, one should expect the model uncertainty to decrease within the support of the training data distribution, that is, on points that are close to the training data in the input space. We will generally call these points *in-distribution* (ID). Conversely, on points that are far away from the training points and thus *out-of-distribution* (OOD), we should not expect the model uncertainty to decrease, since the training data points are not informative enough to make any assertions about the correctness of the given model on these points. In this sense, we would like the uncertainty to be *distance-aware*, that is, it should grow away from the training data (Liu et al., 2020). While the term OOD is used in many different contexts in the literature, we will generally use it to refer to data points that could not have plausibly been generated by the data distribution, as opposed to data points that are just unlikely under the data distribution. For instance, when assessing OOD generalization for a dataset of photos of cats and dogs, OOD points in our nomenclature might be cartoon depictions of cats of dogs. They still can plausibly be classified into the same two classes, but they could not have been generated via the assumed generative process of taking photos of real animals. For the purposes of OOD detection, we might also consider data points that cannot be assigned to any of the existing classes in the training set, for instance, photos of rabbits.

The spectral-normalized Gaussian process (SNGP) (Liu et al., 2020) provides such distance-aware model uncertainties by specifying a Gaussian process prior (Rasmussen & Williams, 2006) over the latent data representations in the penultimate layer of the neural network. Distance-awareness is ensured by using spectral normalization on the pre-logit hidden layers (Behrmann et al., 2019), which encourages bi-Lipschitzness of the mapping from data to latent space. This approximately preserves the distances between data points in the latent representation space by stopping the learned features from collapsing dimensions in the input space onto invariant subspaces (Liu et al., 2020). Note however that while the Euclidean distance in the high-dimensional input space might not always be semantically meaningful, the neural network still has some freedom to transform the data into a feature space where distances are more meaningful, and indeed it has an incentive to do so in order to improve predictive accuracy and uncertainty estimation. Indeed, Liu et al. (2020) showed that this bi-Lipschitzness leads to learning semantically meaningful representations in practice. Note also that this approach only partially captures the model uncertainty, namely in form of uncertainty over the latents. It **does not however capture the uncertainty over the model parameters**, such as for instance Bayesian neural networks (MacKay, 1992; Neal, 1993) or ensemble methods (Lakshminarayanan et al., 2017). This motivates our HetSNGP Ensemble (presented later in the paper, see Section 5.5).

### 2.2 Data uncertainty and the heteroscedastic method

As opposed to the model uncertainty described above, data uncertainty is intrinsic in the data, hence irreducible with more training data. In the case of continuous data (e.g., regression problems), data uncertainty often comes in the form of random noise on the measurements. For discrete data (e.g., classification problems), it usually arises as incorrectly labelled samples. This label noise can be class- and input-dependent (Beyer et al., 2020). For instance, the ImageNet dataset (Deng et al., 2009), contains 100 different classes of dog breeds, which are often hard for human labelers to distinguish from each other. Modeling this type of data uncertainty can improve the calibration and robustness of predictive models (Collier et al., 2021).

A model that does explicitly handle the input- and class-dependent label noise is the heteroscedastic method (Collier et al., 2021). The heteroscedastic method models input- and class-dependent noise by introducing a latent multivariate Gaussian distribution on the softmax logits of a standard neural network classifier. The covariance matrix of this latent distribution is a function of the input (heteroscedastic) and models inter-class correlations in the logit noise.

## 3 Method

### 3.1 Setup and Notation

Let us consider a dataset $\mathcal{D} = \{(\boldsymbol{x}_i, y_i)\}_{i=1}^N$ of input-output pairs, where $\boldsymbol{x}_i \in \mathbb{R}^d$ and $y_i \in \{1, \ldots, K\}$, that is, a classification problem with $K$ classes. The data examples are assumed to be sampled *i.i.d.* from some true data-generating distribution as $(\boldsymbol{x}_i, y_i) \sim p^*(\boldsymbol{x}, y)$.

### 3.2 Generative process

To jointly model the two different types of uncertainty, we propose a hierarchical model of two latent random variables, which we denote by $\boldsymbol{f}$ and $\boldsymbol{u}$. $\boldsymbol{f}$ is a latent function value associated to the input $\boldsymbol{x}$ (as in the Gaussian process literature) and is designed to capture the model uncertainty, while $\boldsymbol{u}$ is a latent vector of logits (or *utilities*) that capture the data uncertainty, similar to the setup in Collier et al. (2020), which was inspired by the econometrics literature (Train, 2009). Similarly to Liu et al. (2020), we place a latent Gaussian process (GP) prior over $\boldsymbol{f}$, as $p(\boldsymbol{f}) = \mathcal{GP}(0, k_\theta(\cdot, \cdot))$, where $k_\theta$ is a parameterized kernel function with parameters $\theta$. Note that, in our case, this kernel is parameterized by the learned neural network, so it allows for an expressive feature mapping that can learn to represent rich semantic features in the data. Evaluating this kernel on all pairwise combinations of data points in $\boldsymbol{x}$ yields the kernel matrix $\boldsymbol{K}_\theta(\boldsymbol{x}, \boldsymbol{x})$. We then define $\boldsymbol{u}$ as a (heteroscedastically) noisy observation of $\boldsymbol{f}$.

Stacking the variables across the whole dataset gives us the matrices $\boldsymbol{F}, \boldsymbol{U} \in \mathbb{R}^{N \times K}$. We refer to their respective rows as $\boldsymbol{f}_i, \boldsymbol{u}_i \in \mathbb{R}^K$ and their columns as $\boldsymbol{f}_c, \boldsymbol{u}_c \in \mathbb{R}^N$. The columns are independent under the GP prior, but the rows are not. Conversely, the columns are correlated in the heteroscedastic noise model, while the rows are independent. A hierarchical model using both uncertainties therefore leads to logits that are correlated across both data points and classes. While this joint modeling of model and data uncertainty will not necessarily be useful on all tasks (e.g., in-distribution prediction on data with homoscedastic noise), we expect it to lead to improvements on out-of-distribution inputs from datasets with class-dependent label noise. We will see in Section 5 that this is actually the case in practically relevant settings, such as when training on Imagenet-21k and predicting on Imagenet. We now give a formal description of the model.

We assume the full generative process is

$$\boldsymbol{f}_c \sim \mathcal{N}(\boldsymbol{0}, \boldsymbol{K}_\theta(\boldsymbol{x}, \boldsymbol{x})) \tag{1}$$

$$\boldsymbol{u}_i \sim \mathcal{N}(\boldsymbol{f}_i, \boldsymbol{\Sigma}(\boldsymbol{x}_i; \varphi)) \tag{2}$$

$$p(y_i = c \,|\, \boldsymbol{u}_i) = \mathbb{1}\left[c = \arg\max_k u_{ik}\right] \tag{3}$$

We can compute the marginal distribution, that is $p(y \,|\, \boldsymbol{x}) = \mathbb{E}_{\boldsymbol{u}}[p(y \,|\, \boldsymbol{u})] = \int p(y \,|\, \boldsymbol{u})\, p(\boldsymbol{u} \,|\, \boldsymbol{x})\, d\boldsymbol{u}$. Intuitively, $\boldsymbol{f}$ captures the model uncertainty, that is, the uncertainty about the functional mapping between $\boldsymbol{x}$ and $y$ on the level of the latents. It uses the covariance between data points to achieve this distance-awareness, namely it uses the kernel function to assess the similarity between data points, yielding an uncertainty estimate that grows away from the data. On the other hand, $\boldsymbol{u}$ captures the data uncertainty, by explicitly modeling the per-class uncertainty on the level of the logits. $\varphi$ in (2) can learn to encode correlations in the noise between different classes (e.g., the dog breeds in ImageNet). It does not itself capture the model uncertainty, but inherits it through its hierarchical dependence on $\boldsymbol{f}$, such that the resulting probability $p(y \,|\, \boldsymbol{u})$ ultimately jointly captures both types of uncertainty.

In practice, we usually learn the kernel using deep kernel learning (Wilson et al., 2016), that is, we define it as the RBF kernel $k_\theta(\boldsymbol{x}_i, \boldsymbol{x}_j) = k_{RBF}(\boldsymbol{h}_i, \boldsymbol{h}_j) = \exp(\|\boldsymbol{h}_i - \boldsymbol{h}_j\|_2^2 / \lambda)$ with length scale $\lambda$ and $\boldsymbol{h}_i = h(\boldsymbol{x}_i; \theta) \in \mathbb{R}^m$ with $h(\cdot; \theta)$ being a neural network model (e.g., a ResNet) parameterized by $\theta$. This kernel is then shared between classes. Moreover, following Liu et al. (2020), we typically encourage bi-Lipschitzness of $h$ using spectral normalization (Behrmann et al., 2019), which then leads to an approximate preservation of distances between the input space and latent space, thus allowing for distance-aware uncertainty modeling in the latent space. Additionally, $\boldsymbol{\Sigma}(\cdot; \varphi)$ is usually also a neural network parameterized by $\varphi$. To ease notation, we will typically drop these parameters in the following. To make the model more parameter-efficient, we will in practice use $\boldsymbol{h}_i$ as inputs for $\boldsymbol{\Sigma}(\cdot)$, so that the network parameterizing the GP kernel can share its learned features with the network parameterizing the heteroscedastic noise covariance. This usually means that we can make $\boldsymbol{\Sigma}(\cdot)$ rather small (e.g., just a single hidden layer). Note that the prior over $\boldsymbol{f}_c$ (and thus also $\boldsymbol{u}_c$) has zero mean, thus leading to a uniform output distribution away from the data. This is also reminiscent of multi-task GPs (Williams et al., 2007), where separate kernels are used to model covariances between data points and tasks and are then combined into a Kronecker structure.

### 3.3 Computational approximations

The generative process above requires computing two integrals (i.e., over $\boldsymbol{u}$ and $\boldsymbol{f}$) and inferring an exact GP posterior, making it computationally intractable. We make several approximations to ensure tractability.

**Low-rank approximation.** The heteroscedastic covariance matrix $\boldsymbol{\Sigma}(\boldsymbol{x}_i; \varphi)$ is a $K \times K$ matrix which is a function of the input. Parameterizing this full matrix is costly in terms of computation, parameter count, and memory. Therefore, following Collier et al. (2021), we make a low-rank approximation $\boldsymbol{\Sigma}(\boldsymbol{x}_i; \varphi) = \boldsymbol{V}(\boldsymbol{x}_i)\boldsymbol{V}(\boldsymbol{x}_i)^\top + \boldsymbol{d}^2(\boldsymbol{x}_i)\boldsymbol{I}_K$. Here, $\boldsymbol{V}(\boldsymbol{x}_i)$ is a $K \times R$ matrix with $R \ll K$ and $\boldsymbol{d}^2(\boldsymbol{x}_i)$ is a $K$ dimensional vector added to the diagonal of $\boldsymbol{V}(\boldsymbol{x}_i)\boldsymbol{V}(\boldsymbol{x}_i)^\top$.

Also following Collier et al. (2021), we introduce a parameter-efficient version of our method to enable scaling to problems with many classes, for instance, ImageNet21k which has 21,843 classes. In the standard version

of our method, $\boldsymbol{V}(\boldsymbol{x}_i)$ is parameterized as an affine transformation of the shared representation $\boldsymbol{h}_i$. In this way, HetSNGP can be added as a single output layer on top of a base network. For the parameter-efficient HetSNGP, we parameterize $\boldsymbol{V}(\boldsymbol{x}_i) = \boldsymbol{v}(\boldsymbol{x}_i)\boldsymbol{1}_R^\top \odot \boldsymbol{V}$ where $\boldsymbol{v}(\boldsymbol{x}_i)$ is a vector of dimension $K$, $\boldsymbol{1}_R$ is a vector of ones of dimension $R$, $\boldsymbol{V}$ is a $K \times R$ matrix of free parameters, and $\odot$ denotes element-wise multiplication. This particular choice of approximation has shown a favorable tradeoff between performance and computational cost in the experiments conducted by Collier et al. (2021), but naturally other approximations would be possible. For instance, one could use a Kronecker-factored (KFAC) approximation (Martens & Grosse, 2015; Botev et al., 2017) or a k-tied Gaussian parameterization (Swiatkowski et al., 2020). Exploring these different options could be an interesting avenue for future work.

**Random Fourier feature approximation.** Computing the exact GP posterior requires $\mathcal{O}(N^3)$ operations (because ones needs to invert an $N \times N$ kernel matrix), which can be prohibitive for large datasets. Following Liu et al. (2020), we thus use a random Fourier feature (RFF) approximation (Rahimi & Recht, 2007), leading to a low-rank approximation of the kernel matrix as $\boldsymbol{K}_\theta(\boldsymbol{x}, \boldsymbol{x}) = \boldsymbol{\Phi}\boldsymbol{\Phi}^\top$ ($\boldsymbol{\Phi} \in \mathbb{R}^{N \times m}$), with random features $\boldsymbol{\Phi}_i = \sqrt{\frac{2}{m}}\cos(\boldsymbol{W}\boldsymbol{h}_i + \boldsymbol{b})$ where $\boldsymbol{W}$ is a fixed weight matrix with entries sampled from $\mathcal{N}(0, 1)$ and $\boldsymbol{b}$ is a fixed bias vector with entries sampled from $\mathcal{U}(0, 2\pi)$. This approximates the infinite-dimensional reproducing kernel Hilbert space (RKHS) of the RBF kernel with a subspace spanned by $m$ randomly sampled basis functions. It reduces the GP inference complexity to $\mathcal{O}(Nm^2)$, where $m$ is the dimensionality of the latent space. We can then write the model as a linear model in this feature space, namely

$$\boldsymbol{u}_i = \boldsymbol{f}_i + \boldsymbol{d}(\boldsymbol{x}_i) \odot \boldsymbol{\epsilon}_K + \boldsymbol{V}(\boldsymbol{x}_i)\boldsymbol{\epsilon}_R \quad \text{with} \quad \boldsymbol{f}_c = \boldsymbol{\Phi}\boldsymbol{\beta}_c \tag{4}$$
$$\text{and} \quad p(\boldsymbol{\beta}_c) = \mathcal{N}(\boldsymbol{0}, \boldsymbol{I}_m); \; p(\boldsymbol{\epsilon}_K) = \mathcal{N}(\boldsymbol{0}, \boldsymbol{I}_K); \; p(\boldsymbol{\epsilon}_R) = \mathcal{N}(\boldsymbol{0}, \boldsymbol{I}_R)$$

Here, $\boldsymbol{\Phi}\boldsymbol{\beta}_c$ is a linear regressor in the space of the random Fourier features and the other two terms of $\boldsymbol{u}_i$ are the low-rank approximation to the heteroscedastic uncertainties (as described above). Again, $m$ can be tuned as a hyperparameter to trade off computational accuracy with fidelity of the model. Since most datasets contain many redundant data points, one often finds an $m \ll N$ that models the similarities in the data faithfully.

**Laplace approximation.** When using a Gaussian likelihood (i.e., in a regression setting), the GP posterior inference can be performed in closed form. However, for classification problems this is not possible, because the Categorical likelihood used is not conjugate to the Gaussian prior. We thus need to resort to approximate posterior inference. Again following Liu et al. (2020), we perform a Laplace approximation (Rasmussen & Williams, 2006) to the RFF-GP posterior, which yields the closed-form approximate posterior for $\boldsymbol{\beta}_c$

$$p(\boldsymbol{\beta}_c \,|\, \mathcal{D}) = \mathcal{N}(\hat{\boldsymbol{\beta}}_c, \widehat{\boldsymbol{\Sigma}}_c) \quad \text{with} \quad \widehat{\boldsymbol{\Sigma}}_c^{-1} = \boldsymbol{I}_m + \sum_{i=1}^N p_{i,c}(1 - p_{i,c})\boldsymbol{\Phi}_i\boldsymbol{\Phi}_i^\top \tag{5}$$

where $p_{i,c}$ is shorthand for the softmax output $p(y_i = c \,|\, \hat{\boldsymbol{u}}_i)$ as defined in Eq. (3), where $\hat{u}_{i,c} = \boldsymbol{\Phi}_i^\top \hat{\boldsymbol{\beta}}_c$. The derivation of this is deferred to Appendix A.2. Here, $\widehat{\boldsymbol{\Sigma}}_c^{-1}$ can be cheaply computed over minibatches of data by virtue of being a sum over data points. Moreover, $\hat{\boldsymbol{\beta}}_c$ is the MAP solution, which can be obtained using gradient descent on the unnormalized log posterior, $-\log p(\boldsymbol{\beta} \,|\, \mathcal{D}) \propto -\log p(\mathcal{D} \,|\, \boldsymbol{\beta}) - \|\boldsymbol{\beta}\|^2$, where the squared norm regularizer stems from the standard Gaussian prior on $\boldsymbol{\beta}$. Using our likelihood, the training objective takes the form of a ridge-regularized cross-entropy loss. We also use this objective to train the other trainable parameters $\theta$ and $\varphi$ of our model.

Note that this approximation is necessarily ignoring multi-modality of the posterior, since it can by definition only fit one mode around the MAP solution. Nonetheless, recent works have shown that it can achieve suprisingly good practical performance in many settings and offers a good tradeoff between performance and computational cost (Immer et al., 2021b;a).

**Monte Carlo approximation.** Finally, we are ultimately interested in $\mathbb{E}_{\boldsymbol{u}}[p(y \,|\, \boldsymbol{u})] = \int p(y \,|\, \boldsymbol{u})\, p(\boldsymbol{u} \,|\, \boldsymbol{x})\, d\boldsymbol{u}$, which again would require solving a high-dimensional integral and is thus computationally intractable. Following Collier et al. (2020), we approximate it using Monte Carlo samples.

Moreover, we approximate the argmax in Eq. (3) with a softmax. To control the trade-off between bias and variance in this approximation, we introduce an additional temperature parameter $\tau$ into this softmax, leading to

$$p(y_i = c \,|\, \boldsymbol{x}_i) = \frac{1}{S} \sum_{s=1}^{S} p(y_i = c \,|\, \boldsymbol{u}_i^s) = \frac{1}{S} \sum_{s=1}^{S} \frac{\exp(u_{i,c}^s/\tau)}{\sum_{k=1}^{K} \exp(u_{i,k}^s/\tau)} \tag{6}$$
$$\text{with} \quad u_{i,c}^s = \boldsymbol{\Phi}_i^\top \boldsymbol{\beta}_c^s + \boldsymbol{d}(\boldsymbol{x}_i) \odot \boldsymbol{\epsilon}_K^s + \boldsymbol{V}(\boldsymbol{x}_i)\boldsymbol{\epsilon}_R^s$$
$$\text{and} \quad \boldsymbol{\beta}_c^s \sim p(\boldsymbol{\beta}_c \,|\, \mathcal{D}) \quad \text{and} \quad \boldsymbol{\epsilon}_K^s \sim \mathcal{N}(\boldsymbol{0}, \boldsymbol{I}_K) \quad \text{and} \quad \boldsymbol{\epsilon}_R^s \sim \mathcal{N}(\boldsymbol{0}, \boldsymbol{I}_R)$$

It should be noted that we only use Eq. (6) at test time, while during training we replace $\boldsymbol{\beta}_c^s$ with its mean $\hat{\boldsymbol{\beta}}_c$ instead of sampling it (see Section 3.4). Note also that, while a larger number $S$ of Monte Carlo samples increases the computational cost, these samples can generally be computed in parallel on modern hardware, such that the wallclock runtime stays roughly the same. Again, the number of samples $S$ can be tuned as a hyperparameter, such that this approximation can be made arbitrarily exact with increased runtime (or, as mentioned, increased parallel computational power).

### 3.4 Implementation

---
**Algorithm 1** HetSNGP training
---
**Require:** dataset $\mathcal{D} = \{(\boldsymbol{x}_i, y_i)\}_{i=1}^{N}$
  Initialize $\theta, \varphi, \boldsymbol{W}, \boldsymbol{b}, \widehat{\boldsymbol{\Sigma}}, \hat{\boldsymbol{\beta}}$
  **for** train_step $= 1$ to max_step **do**
    Take minibatch $(\boldsymbol{X}_i, \boldsymbol{y}_i)$ from $\mathcal{D}$
    **for** $s = 1$ to $S$ **do**
      $\epsilon_K^s \sim \mathcal{N}(\boldsymbol{0}, \boldsymbol{I}_K), \epsilon_R^s \sim \mathcal{N}(\boldsymbol{0}, \boldsymbol{I}_R)$
      $u_{i,c}^s = \boldsymbol{\Phi}_i^\top \hat{\boldsymbol{\beta}}_c + \boldsymbol{d}(\boldsymbol{x}_i) \odot \boldsymbol{\epsilon}_K^s + \boldsymbol{V}(\boldsymbol{x}_i)\boldsymbol{\epsilon}_R^s$
    **end for**
    $\mathcal{L} = -\frac{1}{S} \sum_{s=1}^{S} \log p(\boldsymbol{X}_i, \boldsymbol{y}_i \,|\, \boldsymbol{u}^s) - \|\hat{\boldsymbol{\beta}}\|^2$
    Update $\{\theta, \varphi, \hat{\boldsymbol{\beta}}\}$ via SGD on $\mathcal{L}$
    **if** final_epoch **then**
      Compute $\{\widehat{\boldsymbol{\Sigma}}_c^{-1}\}_{c=1}^{K}$ as per Eq. (5)
    **end if**
  **end for**

---
**Algorithm 2** HetSNGP prediction
---
**Require:** test example $\boldsymbol{x}^*$
  $\boldsymbol{\Phi}^* = \sqrt{\frac{2}{m}} \cos(\boldsymbol{W}\boldsymbol{h}(\boldsymbol{x}^*) + \boldsymbol{b})$
  **for** $s = 1$ to $S$ **do**
    $\boldsymbol{\beta}_c^s \sim p(\boldsymbol{\beta}_c \,|\, \mathcal{D})$
    $\epsilon_K^s \sim \mathcal{N}(\boldsymbol{0}, \boldsymbol{I}_K), \epsilon_R^s \sim \mathcal{N}(\boldsymbol{0}, \boldsymbol{I}_R)$
    $u_{*,c}^s = \boldsymbol{\Phi}^{*\top} \boldsymbol{\beta}_c^s + \boldsymbol{d}(\boldsymbol{x}^*) \odot \boldsymbol{\epsilon}_K^s + \boldsymbol{V}(\boldsymbol{x}^*)\boldsymbol{\epsilon}_R^s$
  **end for**
  $p(y^* = c \,|\, \boldsymbol{x}^*) = \frac{1}{S} \sum_{s=1}^{S} \frac{\exp(u_{*,c}^s/\tau)}{\sum_{k=1}^{K} \exp(u_{*,k}^s/\tau)}$
  Predict $y^* = \arg\max_c p(y^* = c \,|\, \boldsymbol{x}^*)$

---

The training of our proposed model is described in Algorithm 1 and the prediction in Algorithm 2.

**Intuition.** The distance-aware uncertainties of the SNGP are modeled through $\boldsymbol{\Phi}$, which itself depends on the latent representations $\boldsymbol{h}$, which are approximately distance-preserving thanks to the bi-Lipschitzness. This means that when we have an input $\boldsymbol{x}_{\text{far}}$ that is far away from the training data, the values of the RBF kernel $\exp(\|\boldsymbol{h} - \boldsymbol{h}_{\text{far}}\|_2^2/\lambda)$ will be small, and thus the uncertainty of $\boldsymbol{f}_c$ will be large. The resulting output uncertainties will therefore be large regardless of the heteroscedastic variance $\boldsymbol{\Sigma}(\boldsymbol{x}_i)$ (since they are additive) allowing for effective OOD detection. Conversely, if we have an in-distribution input, the kernel values will be large and the model uncertainty captured in $\boldsymbol{f}_c$ will be small. In this case, the additive output uncertainty will be dominated by the heteroscedastic variance $\boldsymbol{\Sigma}(\boldsymbol{x}_i)$, allowing for effective in-distribution label noise modeling.

While at first it might seem straightforward to combine the two types of uncertainty, it should be noted that the proposed hierarchical model is by no means an arbitrary choice. Rather, it is the only design we found that robustly achieves a disentanglement of the two uncertainty types during training. When using other architectures, for instance, using the heteroscedastic uncertainty directly in the likelihood of the GP or using two additive random variables with respective GP and heteroscedastic distributions, the two

Table 1: Comparison to related work. We see that our proposed HetSNGP model is the only one that captures distance-aware model uncertainties and data uncertainties in a scalable way.

| | Data uncertainty | Model uncertainty | | Distance aware | Scaling |
|---|---|---|---|---|---|
| | | Ensemble (size=M) | approx. Bayesian inference | | |
| Deterministic | ✗ | ✗ | ✗ | ✗ | ✓ |
| Deep ensemble | ✗ | ✓ | ✗ | ✗ | $\mathcal{O}(M)$ |
| SNGP | ✗ | ✗ | ✓ | ✓ | ✓ |
| MIMO | ✗ | ✓ | ✗ | ✗ | ✓ |
| Posterior Network | ✓ | ✗ | ✓ | ✓ | # classes |
| Het. | ✓ | ✗ | ✗ | ✗ | ✓ |
| HetSNGP | ✓ | ✗ | ✓ | ✓ | ✓ |
| HetSNGP Ensemble | ✓ | ✓ | ✓ | ✓ | $\mathcal{O}(M)$ |
| HetSNGP + MIMO | ✓ | ✓ | ✓ | ✓ | ✓ |

models will compete for the explained uncertainty and the heteroscedastic prediction network will typically try to explain as much uncertainty as possible. Only in this hierarchical model setting, where the SNGP adds uncertainty to the mean first and the heteroscedastic uncertainty is added in the second step, can we robustly achieve a correct assignment of the different uncertainties using gradient-based optimization.

**Challenges and limitations.** As outlined above, the main challenge in this model is the inference. We chose an approximate inference scheme that is computationally efficient, using a series of approximations, and modeling the distance-aware and heteroscedastic variances additively. If we wanted to make the model more powerful, at the cost of increased computation, we could thus consider several avenues of improvement: (i) use inducing points or even full GP inference for the GP posterior; (ii) use a more powerful approximation than Laplace, for instance, a variational approximation; or (iii) model the variances jointly, such that we do not only have covariance between data points in $\boldsymbol{f}$ and between classes in $\boldsymbol{u}$, but have a full covariance between different classes of different data points in $\boldsymbol{u}$. All of these ideas are interesting avenues for future research.

## 4 Related Work

**Uncertainty estimation.** There has been a lot of research on uncertainty estimation in recent years (Ovadia et al., 2019), including the early observation that one needs data and model uncertainties for successful use in real-world applications (Kendall & Gal, 2017). There have also been attempts to directly optimize specialized loss functions to improve model uncertainty (Tagasovska & Lopez-Paz, 2019; Pearce et al., 2018). However, if one wants to directly implement prior knowledge regarding the OOD behavior of the model, one usually needs access to OOD samples during training (Yang et al., 2019; Hafner et al., 2020). Another popular idea is to use a hierarchical output distribution, for instance a Dirichlet distribution (Milios et al., 2018; Sensoy et al., 2018; Malinin & Gales, 2018; Malinin et al., 2019; Malinin & Gales, 2019; Hobbhahn et al., 2020; Nandy et al., 2020), such that the model uncertainty can be encoded in the Dirichlet and the data uncertainty in its Categorical distribution samples. This idea was also used in our Posterior Network baseline (Charpentier et al., 2020). While this allows to capture both model and data uncertainty, the model uncertainties are not necessarily distance-aware, which has been shown to be crucial for effective OOD detection (Liu et al., 2020). Moreover, the Dirichlet data uncertainties in these models do not generally allow for correlated class-dependent uncertainties, such as in our model and Collier et al. (2021). Similarly to our idea of learning separate variances, it has also been proposed to treat the output variance variationally and to specify a hierarchical prior over it (Stirn & Knowles, 2020). Finally, single-pass uncertainty approaches such as DUQ (Van Amersfoort et al., 2020) and DUE (van Amersfoort et al., 2021) also capture model uncertainty in a scalable way, but do not additionally capture the data uncertainty. They could possibly be used as a drop-in replacement for the SNGP component in our hierarchical model, which is an interesting avenue for future work.

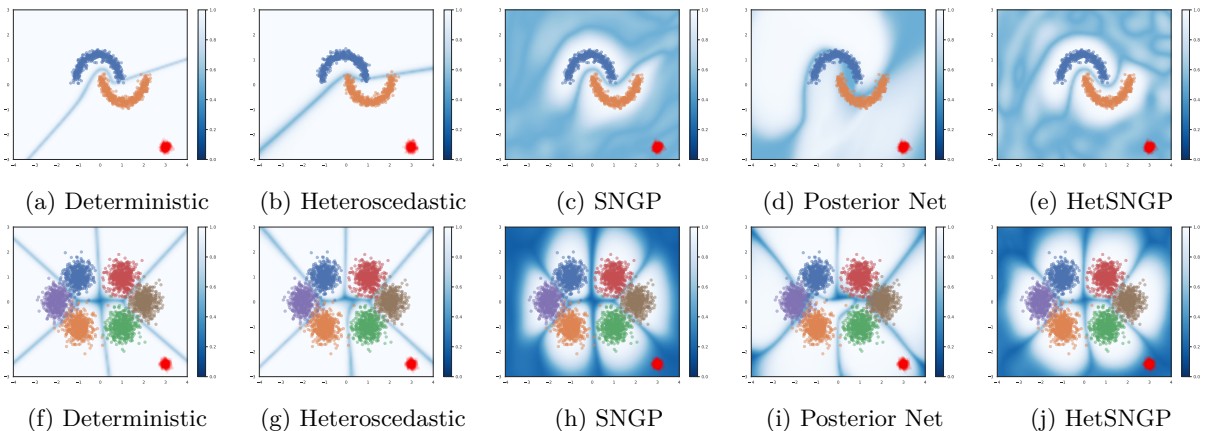

(a) Deterministic  (b) Heteroscedastic  (c) SNGP  (d) Posterior Net  (e) HetSNGP

(f) Deterministic  (g) Heteroscedastic  (h) SNGP  (i) Posterior Net  (j) HetSNGP

Figure 1: Estimated model uncertainties on synthetic datasets. The red cluster of points are unseen OOD points, while the other colors represent different classes of training data. The background color shows the predictive uncertainty (maximum class probability) for the different methods. The deterministic and heteroscedastic methods are overconfident on the OOD data, while the other methods offer distance-aware uncertainties. The uncertainties of the Posterior Network do not grow quickly enough away from the data, such that only the SNGP and proposed HetSNGP provide effective OOD detection.

**Bayesian neural networks and ensembles.** The gold-standard for capturing model uncertainty over the parameters are Bayesian neural networks (Neal, 1993; MacKay, 1992), but they are computationally quite expensive, require a careful choice of prior distribution (Fortuin et al., 2021b; Fortuin, 2021), and require specialized approximate inference schemes, such as Laplace approximation (Immer et al., 2021b;a), variational inference (Hinton & Van Camp, 1993; Graves, 2011; Blundell et al., 2015), or Markov Chain Monte Carlo (Welling & Teh, 2011; Garriga-Alonso & Fortuin, 2021; Fortuin et al., 2021a). A more tractable model class are deep ensemble methods (Lakshminarayanan et al., 2017; Ciosek et al., 2019; D'Angelo et al., 2021; D'Angelo & Fortuin, 2021), although they are computationally still expensive. There are however some ideas to make them less expensive by distilling their uncertainties into simpler models (Malinin et al., 2019; Tran et al., 2020; Havasi et al., 2020; Antorán et al., 2020).

**SNGP and heteroscedastic method.** The models most relevant to our approach are the SNGP (Liu et al., 2020) and the heteroscedastic method (Collier et al., 2020; 2021). SNGP yields distance-aware model uncertainties, which grow away from the training data. The heteroscedastic method models input- and class-dependent label noise data uncertainty inside the training distribution. Heteroscedastic data uncertainty modeling can also improve segmentation models (Monteiro et al., 2020) and has been linked to the cold posterior effect in Bayesian neural networks (Wenzel et al., 2020; Adlam et al., 2020; Aitchison, 2020). Prior work has primarily focused on modeling data and distance-aware model uncertainty separately. But safety-critical practical applications require both types of uncertainty, and we see in Table 1 that our proposed HetSNGP method is the only one which offers joint modeling of distance-aware model uncertainties and data uncertainties, while still being scalable.

## 5 Experiments

We conducted experiments on synthetic data and standard image classification benchmarks. As baselines, we compare against a standard deterministic model (He et al., 2016; Dosovitskiy et al., 2020), the heteroscedastic method (Collier et al., 2021) and SNGP (Liu et al., 2020). We also compare against the Posterior Network model (Charpentier et al., 2020), which also offers distance-aware uncertainties, but it only applies to problems with few classes. Note that our main motivation for the experiments is to assess whether combining the SNGP and heteroscedastic method can successfully demonstrate the complementary benefits of the two methods. Since not all tasks and datasets require both types of uncertainty at the same time, we do not expect our proposed model to outperform the baselines on all tasks. We would however

Figure 2: Synthetic data task with label noise. From left to right: clean labels, noisy labels, deterministic predictions (0.804 accuracy), SNGP predictions (0.841 accuracy), heteroscedastic predictions (0.853 accuracy), and HetSNGP predictions (0.866 accuracy). We see that the proposed method performs on par with the heteroscedastic method and that they both outperform the other baselines thanks to their label noise modeling capabilities.

expect that, depending on the particular task and required uncertainty type, it would generally outperform one of the two baselines. Our non-synthetic experiments are developed within the open source codebases `uncertainty_baselines` (Nado et al., 2021) and `robustness_metrics` (Djolonga et al., 2020) (to assess the OOD performances). Implementation details are deferred to Appendix A.1. Our implementation of the HetSNGP is available as a layer in `edward2` (`https://github.com/google/edward2/blob/main/edward2/tensorflow/layers/hetsngp.py`) and the experiments are implemented in `uncertainty_baselines` (e.g., `https://github.com/google/uncertainty-baselines/blob/main/baselines/imagenet/hetsngp.py`).

To measure the predictive performance of the models, we use their accuracy (*Acc*) and negative log-likelihood (*NLL*). To measure their uncertainty calibration, we use the expected calibration error (*ECE*) (Naeini et al., 2015). To measure OOD detection, we use the area under the receiver-operator-characteristic curve when using the predictive uncertainty as a score (*ROC*) and the false-positive rate (rate of classifying an OOD data point as being in-distribution) at 95% recall (*FPR95*) (Fort et al., 2021).

### 5.1 Synthetic experiments

**Synthetic OOD data**  Following the SNGP paper (Liu et al., 2020), to assess the distance-awareness property of HetSNGP in a visually verifiable way, we performed experiments on two-dimensional synthetic datasets: (i) two moons and (ii) a Gaussian mixture. We see in Fig. 1 that neither the deterministic nor heteroscedastic models offer distance-aware uncertainties and are therefore overconfident on the OOD data (red points). While the Posterior network offers uncertainties that seem mildly distance-aware, on these datasets, those uncertainties grow more slowly when moving away from the training data than in the case of the SNGP methods. As a result, the SNGP methods enable a more effective OOD detection, especially on near-OOD tasks (close to the training data), where the Posterior network seems to be overconfident. Moreover, we will see later that scaling the Posterior network to datasets with many classes is too expensive to be practical. Only the SNGP and our proposed HetSNGP are highly uncertain on the OOD points, thus allowing for effective OOD detection, while at the same time being computationally scalable to larger problems with many more classes.

**Synthetic heteroscedastic data**  We now wish to verify in a low-dimensional setting that HetSNGP also retains the in-distribution heteroscedastic modeling property of the heteroscedastic method. We use the noisy concentric circles from Berthon et al. (2021), where the three circular classes have the same mean, but different amounts of label noise. We see in Fig. 2 that the heteroscedastic model and our proposed method are able to capture this label noise and thus achieve a better performance, while the deterministic baseline and the SNGP are not. The intuition for this advantage is that when capturing the class-dependent label noise, these models can ignore certain noisy data examples and therefore, when in doubt, assign larger predictive probabilities to the classes that are less noisy in the data, which in expectation will lead to more accurate predictions than from the other models, which are overfitting to the label noise. We see this visually in Fig. 2, where the predicted labels for the heteroscedastic and HetSNGP models capture the true geometric structure of the clean labels, while the deterministic and SNGP methods erroneously fit the shape of the noisy labels. Note that Appendix C in Collier et al. (2021) also offers a theoretical explanation for this behavior, using a Taylor series approximation argument. Based on these two synthetic experiments, it becomes apparent

Table 2: Results on CIFAR-100 with Places365 as Far-OOD and CIFAR-10 as Near-OOD datasets. We report means and standard errors over 10 runs. Bold numbers are within one standard error of the best performing model. Our model outperforms the baselines in terms of accuracy on corrupted data and is on par with the best models on OOD detection.

| Method | ↑ID Acc | ↓ID NLL | ↑Corr Acc | ↓Corr NLL | ↑Near-ROC | ↓Near-FPR95 | ↑Far-ROC | ↓Far-FPR95 |
|---|---|---|---|---|---|---|---|---|
| Det. | **0.808** $_{\pm 0.000}$ | 0.794 $_{\pm 0.002}$ | 0.455 $_{\pm 0.001}$ | 2.890 $_{\pm 0.010}$ | **0.506** $_{\pm 0.006}$ | 0.963 $_{\pm 0.005}$ | 0.442 $_{\pm 0.060}$ | 0.935 $_{\pm 0.041}$ |
| Post.Net. | 0.728 $_{\pm 0.001}$ | 1.603 $_{\pm 0.014}$ | 0.444 $_{\pm 0.001}$ | 3.086 $_{\pm 0.009}$ | 0.472 $_{\pm 0.013}$ | 1.000 $_{\pm 0.000}$ | **0.518** $_{\pm 0.016}$ | 1.000 $_{\pm 0.000}$ |
| Het. | 0.807 $_{\pm 0.001}$ | 0.782 $_{\pm 0.002}$ | 0.447 $_{\pm 0.001}$ | 3.130 $_{\pm 0.018}$ | 0.496 $_{\pm 0.005}$ | **0.957** $_{\pm 0.004}$ | 0.420 $_{\pm 0.024}$ | 0.958 $_{\pm 0.010}$ |
| SNGP | 0.797 $_{\pm 0.001}$ | **0.762** $_{\pm 0.002}$ | 0.466 $_{\pm 0.001}$ | **2.339** $_{\pm 0.007}$ | 0.493 $_{\pm 0.011}$ | 0.961 $_{\pm 0.005}$ | **0.518** $_{\pm 0.073}$ | **0.919** $_{\pm 0.030}$ |
| HetSNGP (ours) | 0.799 $_{\pm 0.001}$ | 0.856 $_{\pm 0.003}$ | **0.471** $_{\pm 0.001}$ | 2.565 $_{\pm 0.007}$ | **0.499** $_{\pm 0.010}$ | **0.955** $_{\pm 0.005}$ | **0.525** $_{\pm 0.038}$ | **0.910** $_{\pm 0.024}$ |

Table 3: Results on ImageNet. We used ImageNet-C as near-OOD and ImageNet-A as far-OOD. We report the mean and standard error over 10 runs. Bold numbers are within one standard error of the best model. HetSNGP performs best in terms of accuracy on ImageNet-C and ImageNet-A.

| Method | ↑ID Acc | ↓ID NLL | ↓ID ECE | ↑ImC Acc | ↓ImC NLL | ↓ImC ECE | ↑ImA Acc | ↓ImA NLL | ↓ImA ECE |
|---|---|---|---|---|---|---|---|---|---|
| Det. | 0.759 $_{\pm 0.000}$ | 0.952 $_{\pm 0.001}$ | 0.033 $_{\pm 0.000}$ | 0.419 $_{\pm 0.001}$ | 3.078 $_{\pm 0.007}$ | 0.096 $_{\pm 0.002}$ | 0.006 $_{\pm 0.000}$ | 8.098 $_{\pm 0.018}$ | 0.421 $_{\pm 0.001}$ |
| Het. | **0.771** $_{\pm 0.000}$ | **0.912** $_{\pm 0.001}$ | 0.033 $_{\pm 0.000}$ | 0.424 $_{\pm 0.002}$ | 3.200 $_{\pm 0.014}$ | 0.111 $_{\pm 0.001}$ | 0.010 $_{\pm 0.000}$ | 7.941 $_{\pm 0.014}$ | 0.436 $_{\pm 0.001}$ |
| SNGP | 0.757 $_{\pm 0.000}$ | 0.947 $_{\pm 0.001}$ | **0.014** $_{\pm 0.000}$ | 0.420 $_{\pm 0.001}$ | **2.970** $_{\pm 0.007}$ | **0.046** $_{\pm 0.001}$ | 0.007 $_{\pm 0.000}$ | 7.184 $_{\pm 0.009}$ | **0.356** $_{\pm 0.000}$ |
| HetSNGP (ours) | 0.769 $_{\pm 0.001}$ | 0.927 $_{\pm 0.002}$ | 0.033 $_{\pm 0.000}$ | **0.428** $_{\pm 0.001}$ | 2.997 $_{\pm 0.009}$ | 0.085 $_{\pm 0.001}$ | **0.016** $_{\pm 0.001}$ | **7.113** $_{\pm 0.018}$ | 0.401 $_{\pm 0.001}$ |

that our proposed HetSNGP successfully combines the desirable OOD uncertainties of the SNGP with the heteroscedastic uncertainties on these simple datasets. We now proceed to evaluate these properties on more challenging higher-dimensional datasets.

## 5.2 CIFAR experiment

We start by assessing our method on a real-world image dataset; we trained it on CIFAR-100 and used CIFAR-10 as a near-OOD dataset and Places365 (Zhou et al., 2017) as far-OOD. We measure the OOD detection performance in terms of area under the receiver-operator-characteristic curve (ROC) and false-positive-rate at 95% confidence (FPR95). We also evaluated the methods' generalization performance on corrupted CIFAR-100 (Hendrycks & Dietterich, 2019). In Table 2, we see that HetSNGP performs between the heteroscedastic and SNGP methods in terms of in-distribution accuracy, but outperforms all baselines in accuracy on the corrupted data. Moreover, it performs on par with the best-performing models on both near- and far-OOD detection. This suggests that in-distribution, only the heteroscedastic uncertainty is needed, such that both the heteroscedastic method and our HetSNGP outperform the standard SNGP in terms of accuracy. However, on the corrupted data, which is outside of the training distribution, the SNGP outperforms the heteroscedastic method in terms of accuracy and our HetSNGP outperforms both baselines, since both types of uncertainty are useful in this setting.

Table 4: Additional ImageNet OOD results on ImageNet-R and ImageNet-V2. The reported values are means and standard errors over 10 runs. Bold numbers are within one standard error of the best performing model. Our model outperforms the baselines in terms of accuracy on ImageNet-V2.

| Method | ↑ImR Acc | ↓ImR NLL | ↓ImR ECE | ↑ImV2 Acc | ↓ImV2 NLL | ↓ImV2 ECE |
|---|---|---|---|---|---|---|
| Det. | 0.229 $_{\pm 0.001}$ | 5.907 $_{\pm 0.014}$ | 0.239 $_{\pm 0.001}$ | 0.638 $_{\pm 0.001}$ | 1.598 $_{\pm 0.003}$ | 0.077 $_{\pm 0.001}$ |
| Het. | **0.235** $_{\pm 0.001}$ | 5.761 $_{\pm 0.010}$ | 0.251 $_{\pm 0.001}$ | **0.648** $_{\pm 0.001}$ | 1.581 $_{\pm 0.002}$ | 0.084 $_{\pm 0.001}$ |
| SNGP | 0.230 $_{\pm 0.001}$ | **5.344** $_{\pm 0.009}$ | **0.175** $_{\pm 0.001}$ | 0.637 $_{\pm 0.001}$ | **1.552** $_{\pm 0.001}$ | **0.041** $_{\pm 0.001}$ |
| HetSNGP (ours) | 0.232 $_{\pm 0.001}$ | 5.452 $_{\pm 0.011}$ | 0.225 $_{\pm 0.002}$ | **0.647** $_{\pm 0.001}$ | 1.564 $_{\pm 0.003}$ | 0.080 $_{\pm 0.001}$ |

Table 5: Results on ImageNet-21k. The reported values are means and standard errors over 5 runs. Bold numbers are within one standard error of the best performing model. We use standard ImageNet, ImageNet-C, ImageNet-A, ImageNet-R, and ImageNet-V2 as OOD datasets. HetSNGP outperforms the baselines on all OOD datasets.

| Method | ↑ID prec@1 | ↑Im Acc | ↑ImC Acc | ↑ImA Acc | ↑ImR Acc | ↑ImV2 Acc |
|---|---|---|---|---|---|---|
| Det. | 0.471 ± 0.000 | 0.800 ± 0.000 | 0.603 ± 0.000 | 0.149 ± 0.000 | 0.311 ± 0.000 | 0.694 ± 0.000 |
| Het. | **0.480** ± 0.001 | 0.796 ± 0.002 | 0.590 ± 0.001 | 0.132 ± 0.004 | 0.300 ± 0.006 | 0.687 ± 0.000 |
| SNGP | 0.468 ± 0.001 | 0.799 ± 0.001 | 0.602 ± 0.000 | 0.165 ± 0.003 | 0.328 ± 0.005 | 0.696 ± 0.003 |
| HetSNGP (ours) | 0.477 ± 0.001 | **0.806** ± 0.001 | **0.613** ± 0.003 | **0.172** ± 0.007 | **0.336** ± 0.002 | **0.705** ± 0.001 |

### 5.3 ImageNet experiment

A large-scale dataset with natural label noise and established OOD benchmarks is the ImageNet dataset (Deng et al., 2009; Beyer et al., 2020). The heteroscedastic method has been shown to improve in-distribution performance on ImageNet (Collier et al., 2021). We see in Table 3 that HetSNGP outperforms the SNGP in terms of accuracy and likelihood on the in-distribution ImageNet validation set and performs almost on par with the heteroscedastic model. The slight disadvantage compared to the heteroscedastic model suggests a small trade-off due to the restricted parameterization of the output layer and application of spectral normalization.

However, the true benefits of our model become apparent when looking at the ImageNet OOD datasets (Table 3, right side). Here, we still have the noisy label properties from the original ImageNet dataset, such that heteroscedastic uncertainties are useful, but we are also outside of the training distribution, such that distance-aware model uncertainties become crucial. On ImageNet-C and ImageNet-A, we see that our proposed model makes good use of both of these types of uncertainties and thus manages to outperform all the baselines in terms of accuracy. Additional OOD results on ImageNet-R and ImageNet-V2 are shown in Table 4.

### 5.4 ImageNet-21k

We introduce a *new large-scale OOD benchmark* based on ImageNet-21k. We hope this new benchmark will be of interest to future work in the OOD literature. ImageNet-21k is a larger version of the standard ImageNet dataset used above (Deng et al., 2009). It has over 12.8 million training images and 21,843 classes. Each image can have multiple labels, whereas for standard ImageNet, a single label is given per image. In creating our benchmark, we exploit the unique property of ImageNet-21k that its label space is a superset of the 1000 ImageNet classes (class n04399382 is missing).

Having trained on the large ImageNet-21k training set, we then evaluate the model on the 1,000 ImageNet classes (setting the predictive probability of class n04399382 to zero). Despite now being in a setting where the model is trained on an order of magnitude more data and greater than $21\times$ more classes, we can use the standard ImageNet OOD datasets. This assesses the scalability of our method and the scalability of future OOD methods.

In our experiments, we train a ViT-B/16 (Dosovitskiy et al., 2020) vision transformer model on the ImageNet-21k dataset (see Appendix A.1 for training details). To scale to over 21k output classes, we use the parameter-efficient versions of the heteroscedastic and HetSNGP methods. We found that SNGP and HetSNGP underfit to the data when using the posterior $p(\beta_c | \mathcal{D})$ at test time, so instead we use the posterior mode $\hat{\beta}$ at both training and test time. Additionally, we found spectral normalization was not necessary to preserve distance awareness for the ViT architecture; hence ImageNet-21k experiments are run without spectral normalization.

We see in Table 5 that the parameter-efficient heteroscedastic method has the best in-distribution precision@1. However, its generalization performance to the OOD benchmarks is the weakest of all the methods.

Table 6: Results on ImageNet. Ensemble size set to 4. ImageNet-C is used as the OOD dataset. HetSNGP outperforms the baselines in terms of in-distribution accuracy and on all OOD metrics.

| Method | ↑ID Acc | ↓ID NLL | ↓ID ECE | ↑ImC Acc | ↓ImC NLL | ↓ImC ECE |
|---|---|---|---|---|---|---|
| Det Ensemble | 0.779 | 0.857 | 0.017 | 0.449 | 2.82 | 0.047 |
| Het Ensemble | 0.795 | **0.790** | **0.015** | 0.449 | 2.93 | 0.048 |
| SNGP Ensemble | 0.781 | 0.851 | 0.039 | 0.449 | 2.77 | 0.050 |
| HetSNGP Ensemble (ours) | **0.797** | 0.798 | 0.028 | **0.458** | **2.75** | **0.044** |

Our method recovers almost the full in-distribution performance of the heteroscedastic method, significantly outperforming the deterministic and SNGP methods. Notably, it also clearly outperforms all other methods on the OOD datasets. Similarly to the CIFAR experiment, we thus see again that in-distribution, the heteroscedastic method and our HetSNGP both outperform the SNGP, since OOD uncertainties are not that important there, while on the OOD datasets, the SNGP and our HetSNGP both outperform the heteroscedastic method. Our method therefore achieves the optimal combination.

### 5.5 Ensembling experiment

Deep ensembles are popular methods for model uncertainty estimation due to their simplicity and good performance (Lakshminarayanan et al., 2017). We propose a variant of our HetSNGP method which captures a further source of model uncertainty in the form of parameter uncertainty by a deep ensemble of HetSNGP models. HetSNGP Ensemble is therefore a powerful yet simple method for capturing three major sources of uncertainty: (1) data uncertainty in the labels, (2) model uncertainty in the latent representations, and (3) model uncertainty in the parameters.

We compare the HetSNGP Ensemble to a determinstic ensemble as well as ensembles of heteroscedastic and SNGP models. We see in Table 6 that in the case of an ensemble of size four, HetSNGP Ensemble outperforms the baselines on the ImageNet-C OOD dataset in terms of all metrics, while also outperforming them on the standard in-distribution ImageNet dataset in terms of accuracy. Due to computational constraints, we do not have error bars in this experiment. However, we do not expect them to be much larger than in Table 3.

In the above experiments, we have observed that the model uncertainty in the parameters, captured here by ensembling, appears to be complementary and benefits all approaches. We defer to future work the exploration of more efficient variants of HetSNGP Ensemble. In Appendix A.3, we discuss a promising extension based on MIMO (Havasi et al., 2020) and the advantages it would bring.

## 6   Conclusion

We have proposed a new model, the HetSNGP, that jointly captures distance-aware model uncertainties and heteroscedastic data uncertainties. The HetSNGP allows for a favorable combination of these two complementary types of uncertainty and thus enables effective out-of-distribution detection and generalization as well as in-distribution performance and calibration on different benchmark datasets. Moreover, we have proposed an ensembled version of our method, which additionally captures uncertainty in the model parameters and improves performance even further, and a new large-scale out-of-distribution benchmark based on the ImageNet-21k dataset.

### Acknowledgments

VF acknowledges funding from the Swiss Data Science Center through a PhD Fellowship, as well as from the Swiss National Science Foundation through a Postdoc.Mobility Fellowship, and from St. John's College Cambridge through a Junior Research Fellowship. We thank Shreyas Padhy, Basil Mustafa, Zelda Mariet, and Jasper Snoek for helpful discussions and the anonymous reviewers for valuable feedback on the manuscript.

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

# A  Appendix

## A.1  Implementation details

To assess our proposed model's predictive performance and uncertainty estimation capabilities, we conducted experiments on synthetic two moons data (Pedregosa et al., 2011), a mixture of Gaussians, the CIFAR-100 dataset (Krizhevsky & Hinton, 2009), and the ImageNet dataset (Deng et al., 2009). We compare against a standard deterministic ResNet model as a baseline (He et al., 2016), against the heteroscedastic method (Collier et al., 2020; 2021) and the SNGP (Liu et al., 2020) (which form the basis for our combined model) and against the recently proposed Posterior Network model (Charpentier et al., 2020), which also offers distance-aware uncertainties, similarly to the SNGP. We used the same backbone neural network architecture for all models, which was a fully-connected ResNet for the synthetic data, a WideResNet18 on CIFAR and a ResNet50 in ImageNet.

For most baselines, we used the hyperparameters from the `uncertainty_baselines` library (Nado et al., 2021). On CIFAR, we trained our HetSNGP with a learning rate of 0.1 for 300 epochs and used $R = 6$ factors for the heteroscedastic covariance, a softmax temperature of $\tau = 0.5$ and $S = 5000$ Monte Carlo samples. On ImageNet, we trained with a learning rate of 0.07 for 270 epochs and used $R = 15$ factors, a softmax temperature of $\tau = 1.25$ and $S = 5000$ Monte Carlo samples. We implemented all models in TensorFlow in Python and trained on Tensor Processing Units (TPUs) in the Google Cloud.

We train all Imagnet-21k models for 90 epochs with batch size 1024 on $8 \times 8$ TPU slices. We train using the Adam optimizer with initial learning rate of 0.001 using a linear learning rate decay schedule with termination point 0.00001 and a warm-up period of 10,000 steps. We train using the sigmoid cross-entropy loss function and L2 weight decay with multiplier 0.03. The heteroscedastic method uses a temperature of 0.4, 1,000 Monte Carlo samples and $R = 50$ for the low rank approximation. HetSNGP has the same heteroscedastic hyperparameters except the optimal temperature is 1.5. For SNGP and HetSNGP the GP covariance is approximated using the momentum scheme presented in Liu et al. (2020) with momentum parameter 0.999.

## A.2  Laplace approximation

In this section, we will derive the Laplace posterior in Eq. (5). The derivation follows mostly from the sections 3.4 and 3.5 in Rasmussen & Williams (2006).

First note that the log posterior of $\boldsymbol{\beta}_c$ given the data is

$$\log p(\boldsymbol{\beta}_c \mid \boldsymbol{x}, y) = \log p(y \mid \boldsymbol{\beta}_c) + \log p(\boldsymbol{\beta}_c) - Z \tag{7}$$

where $Z$ is a normalization constant that does not depend on $\boldsymbol{\beta}_c$. Following Rasmussen & Williams (2006), we will denote the unnormalized log posterior as

$$\Psi(\boldsymbol{\beta}_c) = \log p(y \mid \boldsymbol{\beta}_c) + \log p(\boldsymbol{\beta}_c) \tag{8}$$

Recall that the first term is the likelihood and the second term is our prior from Eq. (4).

The Laplace approximation now approximates the posterior with a local second-order expansion around the MAP solution, that is

$$p(\boldsymbol{\beta}_c \mid \boldsymbol{x}, y) \approx \mathcal{N}(\hat{\boldsymbol{\beta}}_c, \boldsymbol{\Lambda}^{-1}) \tag{9}$$

with the MAP solution $\hat{\boldsymbol{\beta}}_c = \arg\max_{\boldsymbol{\beta}_c} \Psi(\boldsymbol{\beta}_c)$ and the Hessian $\boldsymbol{\Lambda} = -\nabla^2 \Psi(\boldsymbol{\beta}_c)|_{\boldsymbol{\beta}_c = \hat{\boldsymbol{\beta}}_c}$.

The MAP solution can be found using standard (stochastic) gradient descent, while the Hessian is given by

$$
\begin{aligned}
\nabla^2 \Psi(\boldsymbol{\beta}_c) &= \nabla^2 \log p(y \mid \boldsymbol{\beta}_c) + \nabla^2 \log p(\boldsymbol{\beta}_c) \\
&= \nabla_{\boldsymbol{\beta}}(\nabla_{\boldsymbol{u}} \log p(y \mid \boldsymbol{u}) \nabla_{\boldsymbol{\beta}} \boldsymbol{u}) - \boldsymbol{I}_m \\
&= \nabla_{\boldsymbol{\beta}}(\nabla_{\boldsymbol{u}} \log p(y \mid \boldsymbol{u}) \boldsymbol{\Phi}) - \boldsymbol{I}_m \\
&= \boldsymbol{\Phi}^\top \nabla_{\boldsymbol{u}}^2 \log p(y \mid \boldsymbol{u}) \boldsymbol{\Phi} - \boldsymbol{I}_m \\
&= -W \boldsymbol{\Phi}^\top \boldsymbol{\Phi} - \boldsymbol{I}_m
\end{aligned}
$$

Table 7: Runtimes of inference per sample for different methods. Runtimes are computed on ImageNet using Resnet50. We see that the methods do not differ strongly.

| METHOD | RUNTIME (MS/EXAMPLE) |
|---|---|
| DETERMINISTIC | 0.04 |
| SNGP | 0.047 |
| MIMO | 0.040 |
| HET. (5000 MC SAMPLES) | 0.061 |
| HETSNGP (100 MC SAMPLES) | 0.045 |

Table 8: MIMO performance on ImageNet (in-dist), ImageNet-C and ImageNet-A. The reported values are means and standard errors over 10 runs.

| ↑ID Acc | ↓ID NLL | ↓ID ECE | ↑ImC Acc | ↓ImC NLL | ↓ImC ECE | ↑ImA Acc | ↓ImA NLL | ↓ImA ECE |
|---|---|---|---|---|---|---|---|---|
| $0.772_{\pm 0.001}$ | $0.901_{\pm 0.004}$ | $0.039_{\pm 0.001}$ | $0.440_{\pm 0.003}$ | $2.979_{\pm 0.003}$ | $0.101_{\pm 0.005}$ | $0.013_{\pm 0.001}$ | $7.777_{\pm 0.042}$ | $0.432_{\pm 0.005}$ |

where we used the chain rule and the fact that $\boldsymbol{u} = \boldsymbol{\Phi}\boldsymbol{\beta}$ and $W$ is a diagonal matrix of point-wise second derivatives of the likelihood, that is, $W_{ii} = -\nabla^2 \log p(y_i \mid \boldsymbol{u}_i)$ (Rasmussen & Williams, 2006). For instance, in the case of the logistic likelihood, $W_{ii} = \boldsymbol{p}_i(1 - \boldsymbol{p}_i)$, where $\boldsymbol{p}_i$ is a vector of output probabilities for logits $\boldsymbol{u}_i$. To get the Hessian at the MAP, we then just need to compute this quantity for $\hat{\boldsymbol{u}} = \boldsymbol{\Phi}\hat{\boldsymbol{\beta}}$.

The approximate posterior is therefore

$$p(\boldsymbol{\beta}_c \mid \boldsymbol{x}, y) \approx \mathcal{N}(\hat{\boldsymbol{\beta}}_c, (W\boldsymbol{\Phi}^T\boldsymbol{\Phi} + \boldsymbol{I}_m)^{-1}) \tag{10}$$

where the precision matrix can be computed over data points (recovering Eq. (5)) as

$$\boldsymbol{\Lambda} = \boldsymbol{I}_m + \sum_{i=1}^{N} \boldsymbol{p}_i(1 - \boldsymbol{p}_i)\boldsymbol{\Phi}_i\boldsymbol{\Phi}_i^{\top} \tag{11}$$

### A.3 Future work: Combination with efficient ensembling

MIMO (Havasi et al., 2020) is a promising efficient ensemble method we could build HetSNGP upon. We tested MIMO alone to assess how promising it is, see Table 8. In Table 1, we summarize the benefits we would have by combining HetSNGP and MIMO. In particular, we hope to preserve the gains of the expensive HetSGNP Ensemble for a fraction of the cost.

### A.4 Runtime comparison

We profiled the runtimes of the different methods and we see in Table 10 on Imagenet-21k that the different methods do not differ strongly in their computational costs. In particular, our HetSNGP performs generally on par with the standard heteroscedastic method.

Table 9: MIMO performance on ImageNet-R and ImageNet-V2. The reported values are means and standard errors over 10 runs.

| ↑ImR Acc | ↓ImR NLL | ↓ImR ECE | ↑ImV2 Acc | ↓ImV2 NLL | ↓ImV2 ECE |
|---|---|---|---|---|---|
| $0.245_{\pm 0.003}$ | $5.851_{\pm 0.039}$ | $0.248_{\pm 0.004}$ | $0.654_{\pm 0.003}$ | $1.538_{\pm 0.010}$ | $0.085_{\pm 0.003}$ |

Table 10: Profiling of different methods on ImageNet-21k using Vision Transformers (B/16). We report the milliseconds and GFLOPS (=$10^9$ FLOPs) per image, both at training and evaluation time. All measurements are made on the same hardware (TPU V3 with 32 cores). We see that the methods do not differ strongly and that the HetSNGP performs on par with the standard heteroscedastic method.

| Model | ms / img (train) | GFLOPS / img (train) | ms / img (eval) | GFLOPS / img (eval) |
|---|---|---|---|---|
| Det | 4.72 | 106.61 | 1.08 | 35.31 |
| SNGP | 4.74 | 106.65 | 1.08 | 35.32 |
| Het | 6.09 | 112.17 | 1.37 | 37.79 |
| HetSNGP | 6.13 | 112.22 | 1.38 | 37.80 |

Table 11: Ablation study with different numbers of MC samples for the HetSNGP on Imagenet. We see that there are no improvements when using more than 100 samples.

| # of MC samples | Accuracy | NLL | ECE |
|---|---|---|---|
| 5000 | 0.763 | 0.961 | 0.041 |
| 1000 | 0.772 | 0.922 | 0.036 |
| 250 | 0.769 | 0.927 | 0.031 |
| 100 | 0.772 | 0.916 | 0.029 |
| 25 | 0.772 | 0.949 | 0.047 |
| 5 | 0.704 | 1.342 | 0.222 |

## A.5 Ablation studies

We also performed several ablation studies to test the effect of the computational approximations in our proposed HetSNGP method. We see in Table 11 that 100 MC samples are already enough to achieve a good performance on Imagenet and that increasing that number beyond 100 does not offer any further improvements. Moreover, we see in Table 12 that the performance does not improve beyond a rank of 7 for the low-rank heteroscedastic covariance matrix and that even a rank of 2 already performs well. What is more, we see in Table 13 that softmax temperatures around 1.0, that is, a standard untempered softmax, perform well in practice. Finally, we see in Table 14 that MAP inference during training is sufficient to perform well and that MC sampling during training does not significantly improve the performance.

Table 12: Ablation study with different ranks for the low-rank heteroscedastic covariance matrix in the HetSNGP on Imagenet. We see that after rank 7, there are no further improvements, and even rank 2 already works well.

| Rank of covariance | Accuracy | NLL | ECE |
|---|---|---|---|
| 2 | 0.772 | 0.910 | 0.032 |
| 7 | 0.774 | 0.907 | 0.035 |
| 15 | 0.772 | 0.922 | 0.036 |
| 30 | 0.767 | 0.941 | 0.033 |
| 50 | 0.766 | 0.928 | 0.034 |

Table 13: Ablation study with different softmax temperatures for the HetSNGP on Imagenet. We see that temperatures around 1.0 (corresponding to a standard softmax) perform well.

| Temperature | Accuracy | NLL | ECE |
|---|---|---|---|
| 0.4 | 0.763 | 0.973 | 0.051 |
| 0.8 | 0.766 | 0.929 | 0.037 |
| 1.25 | 0.772 | 0.922 | 0.036 |
| 1.6 | 0.770 | 0.934 | 0.036 |
| 2.0 | 0.772 | 0.941 | 0.031 |
| 3.0 | 0.767 | 0.983 | 0.026 |
| 5.0 | 0.729 | 1.392 | 0.352 |

Table 14: Ablation study with and without MC sampling for the training of the HetSNGP on Imagenet. We see that MC sampling during training does not significantly improve the performance, so using MAP inference during training is sufficient.

| Method | Accuracy | NLL | ECE |
|---|---|---|---|
| MAP | 0.772 | 0.922 | 0.036 |
| MC sampling | 0.772 | 0.939 | 0.053 |

