# OpenReview forum: "Deep Classifiers with Label Noise Modeling and Distance Awareness"
_TMLR — Accepted by TMLR_

### Review · Reviewer_X4fV · 2022-06-02

**Summary Of Contributions:**

The paper introduces a new model that allows modeling both distance-aware model and data uncertainties. The resulting model, the heteroscedastic spectral-normalized Gaussian process (HetSNGP), extends the spectral-normalized GP (SNGP) by capturing the uncertainty over model parameters. As a result, a fully Bayesian model is obtained.

The presented approach is evaluated on benchmark datasets (including an OOD benchmark). Moreover, the authors propose a new OOD benchmark (ImageNet-21k). The results of the experiments are convincing.


**Broader Impact Concerns:**

The paper does not contain any broader impact statement. However, I do not see any specific concerns.

**Requested Changes:**

1) I would like to see a better motivation for the chosen low-rank approximation. What about other approximations? Why the picked one is better than the others? A (short) discussion would be beneficial.
2) It would be great to see the influence of the temperature in (6) on the final performance. Perhaps, on toy data to get a better insight.
3) I would like to see a comparison between the HetSNGP and the SNGP in terms of computational complexity.

**Strengths And Weaknesses:**

Strengths:
+ A fully Bayesian model, namely, GP + distribution over parameters.
+ Several approximations are proposed to make the model tractable.
+ By using the RBF kernels, the model is distance-aware as expected.
+ The paper is well-written and easy to follow.
+ The experiments are convincing.

Weaknesses:
- Some choices of approximations are not fully motivated.
- The effect of the temperature used in (6) is not fully presented.
- Training could be done with MC samples rather than the MAP solution. I do not see any discussion on that.
- Since the method is an extension of the SNGP, a computational comparison with this method would be beneficial.

---

### Review · Reviewer_QcRU · 2022-06-05

**Summary Of Contributions:**

The paper proposes a latent variable approach to account for both data uncertainty and model uncertainty.  The techincal contribution is a combination of two recent approaches for model uncertainty (Liu et al., 2020) and data uncertainty (Collier et al., 2020, 2021).  The method is evaluated on several large-scale data sets and a new data set, offering improved results over standard training (SGD) and the Liu et al., Collier et al., works.


**Broader Impact Concerns:**

The paper poses no immediate ethical concerns, and no broader impact statement is required.

**Requested Changes:**

I recommend acceptance of the work. The large-scale experiments and new data set will be of interest to the (Bayesian) deep learning community. Here are some minor suggestions which may improve readability and the presentation of the paper:

* Figure 2 is hard to understand for color-blind people -- perhaps a more diverse color scheme can be picked.

* Perhaps a comment on differences between the heavy-tailed nature of the data distribution and "out-of-distribution" can be added in the introduction to clarify the differences.

* The sentence about Laplace approximation being close in performance to true posterior is misleading. It should be removed or rephrased. To best of my knowledge, (Immer et al., 2021a,b) do not compare to the true posterior, which is intractable for deep neural networks.

* In the paragraph before the equations (1), (2), (3) it should be briefly mentioned how the neural networks are used to parametrize f and u. For example, the reader could be pointed to the corresponding part of the appendix.

* Perhaps a table could be added which compares the run-times of "Det." (standard training) with "HetSNGP" under the same hardware setup.  As of now, the method appears expensive (GP inference), and practicioners may question whether the added computation is worth the moderate improvement over standard training.

**Strengths And Weaknesses:**

Strengths:
* The contributions of the paper and differences to recent related works are clearly stated in the introduction and appendix.
* The paper is well written and was easy to understand.
* The new data set and large-scale experiments will be of interest to the community.

Weaknesses:
* This may be a problem with the terminology used in many Bayesian deep learning papers itself, but it is not clear to me what the terms "in-distribution" and "out-of-distribution" really mean. For example, in image classification, it may be difficult to distinguish rare examples from the heavy-tail of the distribution with examples which are truly "out-of-distribution". So even though the examples may be sampled from the same training distribution, they can be substantially different from all the previous training examples. But should we really call this "out-of-distribution" or rather "heavy-tail"?

* In the paper it is written, "After training data has been observed, one should expect the model uncertainty to decrease within the support of the training data distribution, that is, on points that are close to the training data in the input space.".  But for high-dimensional data, it is hard to say when two examples are different or close. The Euclidean distance seems to mostly make sense in the 2D toy examples. For high-dimensional image data such as ImageNet, two images may be very close semantically, but far away in Euclidean distance. Therefore, the Euclidean distance does not make much sense and the use of bi-Lipschitzness of the kernel embedding seems problematic.

* The technical part of the paper is of incremental nature in view of (Liu et al., 2020) and (Collier et al., 2020, 2021).

* There are no ablation studies which investigate the computational approximations. For example, would more accurate approximations improve the performance?

* Both f and u are parametrized using deep neural networks. It was difficult for me to understand whether two different networks are used, or the same one with different "heads".

* While the method obviously scales very well (experiments on ImageNet21k), after reading the experiments, it is still not clear how much computational overhead the method adds over "standard training"?  Is it negligible? Will the code be released?

---

### Review · Reviewer_WKjU · 2022-06-09

**Summary Of Contributions:**

This paper proposes a hierarchical model of two latent random variables (f and u) for input variables and logits to capture the model and data uncertainty respectively. These latent variables are modeled using Gaussian processes as their prior distribution. Their hierarchical modeling allows them to capture the distance aware model and data uncertainties.

**Requested Changes:**

1. Section 3.2: While the authors said that “f is a latent function value associated to the input x”, the kernel of its corresponding Gaussian prior is computed using the hidden representations k_θ(xi , xj ) = k_RBF (hi , hj ) — this makes sense as f is suppose to capture the model uncertainties. However, I feel that this description should be clearly written in the beginning of section 3.2. Otherwise, it can be confusing that “f” should capture the data uncertainty as it is associated with the input variables x.

2. Related works: The authors should move the related work section in the main paper. Since TMLR has a 12 page limit and the current version of the paper contains 10 pages of main context.

3. The authors often referred to the related works without properly comparing their drawbacks/ limitations in the context of their proposed method.  For e.g. in their related work, the authors said that “Another popular idea is to use a hierarchical output distribution, for instance a Dirichlet distribution (Milios et al., 2018; Sensoy et al., 2018; Malinin & Gales, 2018; Malinin et al., 2019; Hobbhahn et al., 2020), such that the model uncertainty can be encoded in the Dirichlet and the data uncertainty in its Categorical distribution samples. This idea was also used in our Posterior Network baseline (Charpentier et al., 2020).”

Also in their experiment section they mentioned that: “We also compare against the Posterior Network model (Charpentier et al., 2020), which also offers distance-aware uncertainties, but it only applies to problems with few classes. ”

My question/ suggested change(s):
a. Evidential models such as Sensoy et al., 2018; Malinin & Gales, 2019; Nandy et al. 2020 do not suffer from this problem. Then why do the authors choose only the posterior network for their comparison (Charpentier et al., 2020)?

b. I also feel that these evidential models are very closely related to the proposed method and should be discussed more carefully. Also, it would be good to have some discussion on models such as DUQ. Finally, please include the missing citations.

(Amersfoort et al. 2020) Uncertainty Estimation Using a Single Deep Deterministic Neural Network, ICML 2020
(Malinin & Gales, 2019) Reverse kl-divergence training of prior networks, NeurIPS 2019
(Nandy et al. 2020) Towards Maximizing the Representation Gap between In-Domain & Out-of-Distribution Examples, NeurIPS 2020

4. Section 4: “Since not all tasks and datasets require both types of uncertainty at the same time, we do not expect our proposed model to outperform the baselines on all tasks.” – I have understood and agreed on this statement. However, I think it would be good to elaborate it up-front before going into the details to section 4.2.

5. Section 4.1 and Figure 1: I think it would be good to bring Figure 1 to page 7 (near section 4.1) for better readability.

6. “ The Posterior network does offer distance-aware uncertainties, which on these datasets grow more slowly away from the training data than the SNGP methods.” :-- I do not understand this statement. In particular, from Figure 1, it seems that SNGP models produced the best results. Perhaps, can you please be more specific/ increase the size of the axes to (-20, 20)*(-20*20)?

7. “We see in Fig. 2 that the heteroscedastic model and our proposed method are able to capture this label noise and thus achieve a better performance, while the deterministic baseline and the SNGP are not.”
I could not understand (or do not agree) with the authors. How capturing label noise (i.e. data uncertainty) leads to better accuracy? In fact, a classifier can produce incorrect predictions due to data-uncertainty (and we may not be able to reduce that as they are irreducible). However, it can still correctly identify the cause of uncertainty as label-noise, providing a better uncertainty estimation. Can you please explain?


**Strengths And Weaknesses:**

Strengths:

1. Hierarchical modeling of model and data uncertainty is a novel and interesting idea.

2. Paper is well-written and easy to read. The proposed model is also well motivated.

Weaknesses: Please refer to the following part.

---

### Decision · Action_Editors · 2022-07-02

**Recommendation:** Accept as is

**Comment:**

The authors propose a Bayesian approach to model simultaneously the model uncertainty and the data uncertainty in deep learning. This is illustrated on an exhaustive set of experiments, with a clear improvement on existing methods.

The reviewers all found the idea interesting, and the experimental results overall convincing and useful for the community. I agree with them. Overall, they also found the paper clear and well written. Each of them pointed out some minor problems: lack of motivation for some choices in the simulations, a few misleading comments, and a few places where the clarity could be improved. They suggested some improvements. The authors submitted a revised version where all these comments were taken into account. The 3 reviewers were satisfied and recommended acceptation of the paper.